# Enhanced geocoding precision for location inference of tweet text using spaCy, Nominatim and Google Maps. A comparative analysis of the influence of data selection

**Helen Ngonidzashe Serere**[1]*, **Bernd Resch**[1,2], **Clemens Rudolf Havas**[1]

**1** Department of Digital and Analytical Sciences, University of Salzburg, Salzburg, Austria, **2** Centre for Geographic Analysis, Harvard University, Cambridge, MA, United States of America

\* helenngonidzashe.serere@plus.ac.at

**Data Availability Statement:** Serere, Helen Ngonidzashe, 2022, "Replication Data for: Analysing the performance of a location inference

## Abstract

Twitter location inference methods are developed with the purpose of increasing the percentage of geotagged tweets by inferring locations on a non-geotagged dataset. For validation of proposed approaches, these location inference methods are developed on a fully geotagged dataset on which the attached Global Navigation Satellite System coordinates are used as ground truth data. Whilst a substantial number of location inference methods have been developed to date, questions arise pertaining the generalizability of the developed location inference models on a non-geotagged dataset. This paper proposes a high precision location inference method for inferring tweets' point of origin based on location mentions within the tweet text. We investigate the influence of data selection by comparing the model performance on two datasets. For the first dataset, we use a proportionate sample of tweet sources of a geotagged dataset. For the second dataset, we use a modelled distribution of tweet sources following a non-geotagged dataset. Our results showed that the distribution of tweet sources influences the performance of location inference models. Using the first dataset we outweighed state-of-the-art location extraction models by inferring 61.9%, 86.1% and 92.1% of the extracted locations within 1 km, 10 km and 50 km radius values, respectively. However, using the second dataset our precision values dropped to 45.3%, 73.1% and 81.0% for the same radius values.

## 1. Introduction

Twitter is one of the most popular social media platforms. With over 500 million tweets generated every day, the platform contains a huge amount of data voluntarily generated across a diverse range of topics. Researchers have capitalised on the large, near real time and diverse nature of Twitter data to address fields such as urban planning [1], emergence response [2, 3], damage assessments [4], movement patterns [5–7] and other related fields. Through these substantiated research, Twitter data has proven to be a reliable and valuable data source, evidenced by the results of the respective studies.

method on various Twitter source distribution",
https://doi.org/10.7910/DVN/LOTEGM, Harvard
Dataverse, V1.

**Funding:** Initials of the authors who received each
award: BR Grant number 878652 Funding Agency:
Austria Research Promotion Agency (FFG) The
funders had no role in study design, data collection
and analysis, decision to publish, or preparation of
the manuscript.

**Competing interests:** The authors have declared
that no competing interests exist.

One major drawback to the use of Twitter data is the rather small percentage of geotagged posts. Of the millions of tweets generated, researchers have found the percentage of geotagged posts to range between 0.35% and 3% of the total tweets generated [8–10]. These rather low percentages of geotagged tweets limit the sample size used for spatial analysis [3] thereby impacting the representativeness of the results.

To alleviate the challenges brought by having lower percentages of geotagged posts, researchers have developed various methods to infer the tweets' location. Depending on the desired precision, researchers have inferred either Twitter user's place of residence [11–13], message location [14, 15], or the user's position at the time of sending the tweet also known as the tweets' point of origin [8, 16, 17].

Location inference methods are often developed and tested on a geotagged dataset where the attached latitude and longitude pairs are used as ground truth data. The current location inference methods are able to infer at most 83% of the tweet's point of origin within a 50 km radius of the ground truth position [9, 16]. Whilst there appears to be progress in accurately inferring Twitter locations, questions still arise as to the possibility to use the developed location inference models on a non-geotagged dataset, which is the ultimate goal of location inference methods. Questions of location inference model transferability essentially emanates from the differences in tweet source distribution between geotagged and non-geotagged datasets, among other factors.

Tweet source refers to the application, through which the tweet was generated. A tweet can either be generated from a native Twitter application such as Twitter for iPad, Twitter for Windows, Twitter for Android etc. or from a third party application such as Instagram, Foursquare swarm, Facebook etc. Geotagged datasets generally contain a higher percentage of tweets generated from third party applications whilst non-geotagged datasets contain a higher percentage of tweets generated from native Twitter applications [10, 18, 19]. The study of [10] found that whilst posts sent via third party applications such as Instagram contain a higher percentage of geotagged tweets, posts sent via native Twitter applications contained approximately 56 times more non-geotagged posts than third party applications. This finding highlights an overrepresentation of third party applications in developed location inference methods and questions the generalizability of developed location inference models to a non-geotagged dataset.

We have two objectives in this paper. First we aim to **enhance the geocoding precision of location inference methods** using available approaches. Second we aim to **investigate the influence of tweet source distribution** when developing location inference models. We answer the stated questions by developing a location inference method and evaluating the performance of the proposed approach on two sample datasets. The first dataset (Dataset A) contains a proportionate sample of tweet source distribution of a geotagged dataset. Given the absence of ground truth data in a non-geotagged dataset, the second dataset (Dataset B) contains geotagged tweets with the tweet source distribution modelled to suit the distribution of tweet sources in a non-geotagged dataset.

## 2. Previous work on location inference methods

The need to increase the percentage of geotagged tweets is not a new topic but one that received a large amount of attention over the last decades. Review work done by [20–22] showed that substantial advancements have been made to infer either the user's place of residence, message location, or the tweet's point of origin (tweet location).

### 2.1 User's home residence

Studies of [13, 23] were amongst the early works of user home location prediction. After the deployment of Twitter in 2006, in 2010 [13] proposed an approach that uses language models

on tweet text to predict a user's home location. In the proposed approach, the authors identified local keywords within the tweet and used the identified keywords to predict the user's home city. They obtained an accuracy of 51% within a 100-mile radius (161 km) of the actual user's profiled home location. In 2011, [23] inferred user home location at a city level using location indicative words within the tweet content. A similar technique was applied in [24] where the authors used language models and machine learning algorithms to infer a user's home country. By evaluating present tweet nouns and google word trends for a set of tweets from each user, the authors were able to geolocate 83% of the users within their home country. A more advanced approach was conducted by [12]. In their research, the authors used all publicly available tweet metadata fields to predict a users' residence at varying levels of granularity. The authors obtained accuracy levels of 61%, 70%, and 80% for city, state and time zone, respectively.

In [25], user social graphs alongside the tweets content were used to infer the user home locations. Their highest F-measure reported from their study was 37.71%. Again using social graphs, [26], analysed user home locations and managed to infer about 80% users within the home city of Dublin. Recently, [11], used the time of opening of a twitter account together with the tweet language to infer the users home location. The authors we able to geolocate 71% of tweets within a country level and 51% within a city level. Although the precision values were measured on a large scale (state, country, time zone), these studies showed the possibilities of inferring a user's residence location using various metadata fields and approaches.

## 2.2 Tweet message location

The message location refers to the location that a user makes reference to. The message location may not necessarily reflect the user's in situ location. In instances such as traffic monitoring or sentiment analysis, researchers are more interested in topics related to specific locations and the user's actual location is not essential. Earlier studies such as those of [27–29] focused on geolocating message locations so as to detect event specific tweets. In 2011, [27] used syntactical information to classify traffic information. Whilst using prepositions the authors were able to determine the start and end location of traffic incidents with an average accuracy of over 90%. In the same year [28] geotagged the points of interests using language and time models whilst [29] inferred message locations from disaster related tweets using a named entity recognition and geocoding. These studies already showed the high possibility of inferring location mentions within tweets.

More recently, [14] used tweet context to identify and classify tweets related to a flooding event. The authors used historically geotagged tweets along with inferred user locations to geolocate the radius of high priority flood victims. Their proposed approach returned a classification accuracy of 81% and a location prediction accuracy of 87%. By utilizing location information contained within a tweet message, [30] analysed user subjectivity and polarity during the 2016 USA elections. They found that extracting inferred location variables significantly increases the accuracy of sentiment and behavioural analysis. In [31], the authors created a hybrid approach that uses Twitter content and user network information to geocode a message location. Their designed method was able to pinpoint the location of event related tweets within a median error of 19 km, 50% of the time. To increase the knowledge of traffic flows, [15], developed a framework to extract the locations of traffic events from tweet text in Greater Mumbai. Their results showed that Twitter users in Greater Mumbai often share traffic information along with location mentions. Assuming that this is the case for other emergencies on a global scale, the need for inferring locations from Twitter data becomes evident.

## 2.3 Tweet's point of origin

In addition to the above discussed location inference methods, the studies of [8, 9, 17, 32–34] predicted users' locations at the time of sending a post (tweet's point of origin). In 2011, [32] used language models to predict a tweet's point of origin from a zip code to Country level. Similarly, [33] used rule generation and identified locations through noun and verb combinations. With this approach, the authors were able to geolocate 79% of the tweets within a country level and 66% within the users' current location.

Using present associations between a location and its relevant keywords, [17] analysed the textual content of a tweet to predict the tweet's point of origin. Using this approach, the authors managed to geolocate 45% of tweets within a radius of 10 km to the tweets' Global Navigation Satellite System (GNSS) position. Additionally, [34] implemented a city level geolocation system to infer the uses' location based on tweet text and user-declared metadata fields using a staking approach. Their approach was able to geolocate 66.5% within a 100 mile (161 km) radius.

Similarly [9] predicted the tweets' point of origin by using a multi-elemental location inference method. They conducted their study on 2,409 flood related English posts generated between Sydney, Australia and surrounding areas. Their approach exploited three location-related elements in the form of the tweet's textual content, profile location and the place labelling. They assigned the finest granular location of the three elements as the inferred tweet location and checked the accuracy by comparing with geotagged tweet coordinates. Their approach resulted in an accuracy of 60% and 83% of tweets within 10 km and 50 km radius, respectively.

There is no doubt that past studies have done a substantial amount of work in improving location inference models. In this paper, we adopt most of the already existing approaches and construct a methodological structure to enhance the precision of content based location inference models.

## 2.4 Incorporating non-geotagged datasets

The primary aim of location inference models is to identify locations in datasets which are not geotagged. A number of studies have incorporated non-geotagged datasets in order to evaluate the efficacy of these models. The research of [35] employed a multi indicator method to investigate the tweets' point of origin. In their method the authors used 1.03 million worldwide geotagged posts and reported an inference accuracy of 22%, 37% and 54% on 1 km, 10km and 50km radius values, respectively. The authors acknowledged the possible bias in using only a geotagged dataset and evaluated their approach on a random selection of 10,000 tweets containing both geotagged and non-geotagged posts. Although no quality assessment could be performed, the authors found similarities in spatial indicators between the two datasets (geotagged and non-geotagged) which suggested that their proposed approach would also perform well on a non-geotagged dataset. Whilst their evaluation was done on a small dataset, their findings suggest possible relatedness between location mentions in a geotagged and a non-geotagged dataset.

While using the user's previously geotagged posts, [17] developed a method that identifies a user's location on a non-geotagged dataset. Their method identifies user locations from previously geotagged posts and learns the associations between each location and its relevant keywords to predict a tweet's point of origin on a non-geotagged dataset. Using this approach, the authors managed to geolocate 45% of tweets within a radius 10 km to the tweets' GNSS position. Their results show the possibility of inferring user locations from a non-geotagged dataset

by associating locations with keywords. However, their approach was limited to inferring locations sent from location related services specifically Foursquare and Japans' Loctouch.

Using a multi-source and multi-model based inference framework, [8] proposed a systematic approach to infer the tweets' origin on a non-geocoded dataset by using Foursquare check in data for Manhattan, New York City, USA. From the Twitter dataset, the authors used 72,601 non-geotagged tweets generated from Facebook and Instagram sources. By geocoding only the extracted locations that coincided with Foursquare points of interests, they geocoded 34% and 44.28% of all tweets within a 250-meter and a 1 km radius, respectively, of a tweet's attached GNSS coordinate. Although their approach returned high accuracy, their results are limited to Foursquare points of interest and do not represent other locations.

Whilst these methods have produced comparative performances, the methods have been limited to either a representation of a geotagged dataset or a selection of tweets from location related services such as Foursquare. Given that the ultimate goal of developing location inference methods is to geotag a non-geotagged dataset, we investigate how differences in source distribution would influence the performances of location inference methods.

## 3. Methods

Fig 1 shows our overall workflow. We start by selecting two data samples. The first dataset (Dataset A) consists of 1% sample of each tweet source strata. The second dataset (Dataset B) contains geotagged tweets which were strategically selected to model the distribution of a non-geocoded Twitter dataset according to the findings of [10].

We perform similar steps for the two datasets. We start by pre-processing the sample tweets. Next, we use a pretrained spaCy model to extract the inferred location entities from the pre-processed tweets. As a third step we geocode the extracted location entities. Owing to the pros and cons of different geocoders, we use and compare the geocoding capabilities of Open-StreetMap's Nominatim geocoder and Google Maps geocoding API. We check the accuracy of the geocoded locations by computing displacement values between the tweets' coordinate tagged point (GNSS position) and the inferred coordinates. Finally, we generate statistical results by combining location entities into groups.

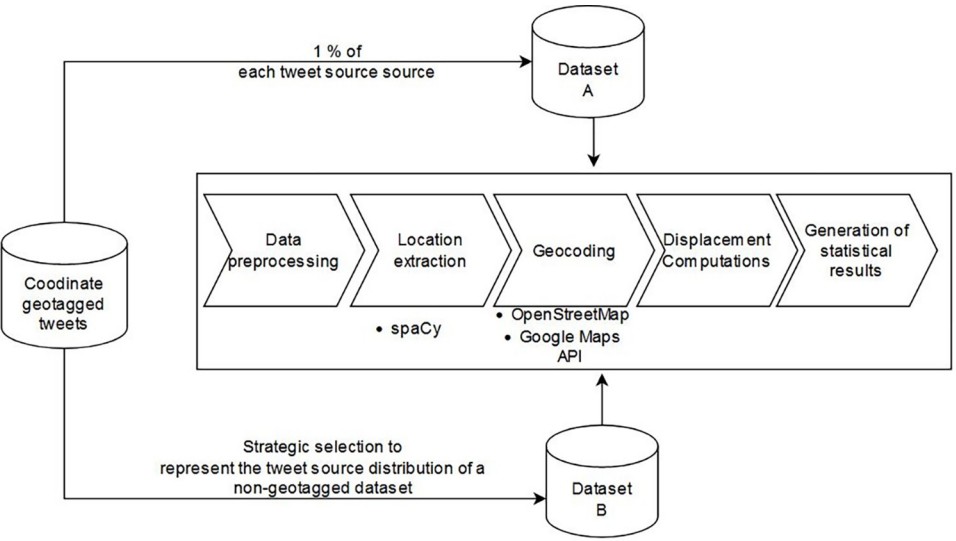

**Fig 1. Overall workflow of the proposed approach.**

In section 3.1 we provide a descriptive overview of the spaCy model used for named entity extraction. In section 3.2. We describe the contents of the two datasets used in the analysis. From section 3.3 to section 3.8 we provide, in a chronological order, a detailed outline of each step of the workflow presented in Fig 1.

### 3.1 spaCy

spaCy is a free, open-source library for advanced Natural Language Processing (NLP) in Python [36, 37]. spaCy supports over 64 language models trained on either newspaper articles, media, blogs, comments etc. spaCy's pretrained English model has four pretrained models; small (sm), medium (md), large (lg) and transformer (trf). The models are distinguished by the size of the training dataset with training data size increasing from the sm to the trf model. For this study, we used spaCy version 3.0 trf pretrained high accuracy English model formally abbreviated as **en_core_web_trf**. This model was trained on 30K web pages, making it a decent fit when applied to a non-structured dataset such as tweets.

The Named Entity Recognition (NER) tool within spaCy extracts entities by entity prediction. This means that, instead of matching place names against a gazetteer, the model uses the text's syntax to predict the probability of a word being a named entity. For example, in the sentence "*I have to Google why Google has a lot of employees*", the model would correctly recognize the first Google as a verb and the latter as an organisation entity, despite being constructed in a similar manner. Because of the models' reliance on sentence syntax for entity extraction [37], it is able to extract entities regardless of spelling errors or presence of noisy elements within the entity. In our previous study [16] we showed that the syntax reliance of spaCy gives it a much higher recall as compared to the gazetteer based DBpedia tool. However, factors such as incorrect punctuation or syntax errors affects the model's performance.

### 3.2 Dataset

For ground truthing, we restricted our analysis to tweets with an attached GNSS position (geotagged tweets). Our dataset contained 16,347,823 geotagged tweets sent between August 2019 and April 2020. The tweets had a GNSS position falling within a bounding box that enclosed the United States of America excluding the states of Hawaii, Alaska and USA island states. The tweets were sent by 492,874 unique users and generated from 739 Twitter sources.

### 3.3 Sample selection

We selected eight tweet sources for the analysis. Our selection was based on the popularity of the tweet source field amongst users. That is, by counting the number of distinct users per Twitter source, we eliminated all twitter sources with less than 1,000 distinct users.

The '*Total tweets*' column in Table 1 shows the number of tweets within our dataset per tweet source. Overall, tweets generated from third party applications (Instagram, Foursquare, Untapped) represented a larger percentage (>97%) of tweets in our dataset. This is in line with the findings of [10, 18, 19, 38], who found larger percentages of geotagged tweets to be generated from third-party applications (Instagram, Foursquare) than from native Twitter applications (Twitter for Android, Twitter for Windows). *Dataset A* consists of a 1% sample selection of each tweet source. Dataset B consists of a strategically selected dataset that models the tweet source distribution of a non-geotagged dataset following the findings of [10].

In Fig 2 we show the geographical distribution of the two sample datasets. Overall, our two datasets were similarly distributed across the study area which suggests similar usage of tweet sources across the USA.

**Table 1. Number of geotagged tweets per tweet source for the full dataset (Total tweets) and the two selected datasets, Dataset A and Dataset B.** The table is ordered by descending order of the Total tweets column.

| Twitter Source | Total tweets | Dataset A | Dataset B |
|---|---|---|---|
| Instagram | 11,556,500 (86.5%) | 115,565 (86.5%) | 1,207 (0.90%) |
| Foursquare | 891,834 (6.68%) | 8,918 (6.68%) | 1,000 (0.75%) |
| Untapped | 472,396 (3.54%) | 4,724 (3.54%) | 1,000 (0.75%) |
| Tweetbot for IOS | 132,663 (1.00%) | 1,327 (1.00%) | 4,231 (3.17%) |
| Hootsuite | 102,197 (0.77%) | 1,022 (0.77%) | 1,000 (0.75%) |
| Foursquare Swarm | 77,969 (0.58%) | 780 (0.58%) | 1,000 (0.75%) |
| Twitter for IPhone | 71,028 (0.53%) | 710 (0.53%) | 71,028 (53.17%) |
| Twitter for Android | 53,121 (0.40%) | 531 (0.40%) | 53,121 (39.77%) |
| Total | 13,357,708 (100%) | 133,577 (100%) | 133,587 (100%) |

## 3.4 Preprocessing

**3.4.1 Text cleaning.** Social media posts are short, unstructured and prone to noise and redundancy [2]. We performed the following preprocessing steps in order to increase model performance by reducing the noise in generated posts. First, we dropped web addresses and emoji characters as the adopted spaCy model could not extract any location information from these characters. We then trimmed off all occurring white spaces (leading, trailing or within the text) and new lines. Next we discarded all posts with less than three words as such short posts cannot be disambiguated in location extraction. Finally we dropped duplicate tweet text so as to avoid any bias due to overrepresentation of tweet. Overall, we tried to maintain sentence syntaxes so as to avoid ambiguity and thus allowing for improved accuracy when extracting named entities.

**3.4.2 Remote location filtering.** Locations inferred within tweet text may either refer to the user's position at the time of sending the post (tweet's point of origin) or to a location outside of the user's current position (remote location). Since our aim was to evaluate the performance of location inference models by computing displacement values from the tweets' GNSS position, we needed only tweets inferring the tweet's point of origin. Thus, we had to filter out all tweets with plausible remote location mentions. Previous studies have filtered remote locations using either textual pattern matching [17] or a combination of local keywords and parts of speech tagging [8]. We performed our location filtering process in two stages. First, using a list of predictive keyword phrases. Second, using temporal Information matching.

*3.4.2.1 Keyword filtering.* Keywords are used on various accounts to retrieve relevant data [39, 40]. Oftentimes, researchers generate a list of keywords matching specific topics of

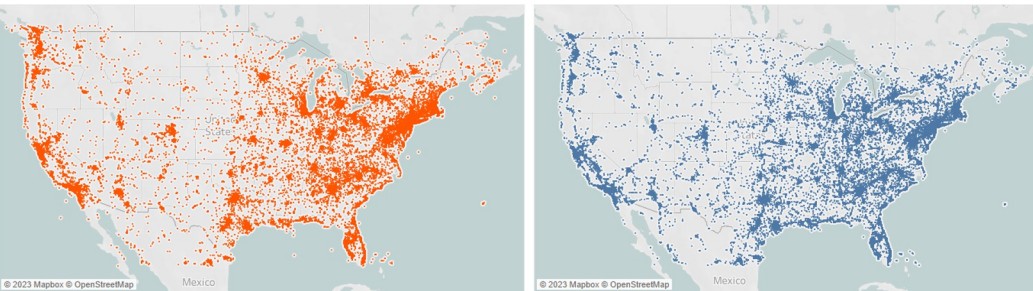

**Fig 2.** Distribution of posts within Dataset A (left) and Dataset B (right). Visually, the posts have a comparatively similar distribution across the study area.

interest. In our case, however, we could not create a list of keywords that return posts with the current user's position. This is because there are numerous ways in which users can refer to their current location. For example; '*standing right in front of the Eiffel Tower*'; '*chilling at McDonalds in Fourth Street, Manchester*'; '*taking a break at Laz Café.* ' etc. Hence, instead of using keywords to extract current user locations, we used keywords to eliminate posts that most likely contained mentions of a users' remote location. We generated, to the best of our ability, an unambiguous list of keyword phrases such as "relocating to", "flying from", "originally from", etc. to filter out posts with possible remote location mentions (see. the S1 Appendix for the full list of keyword phrases). Because not all remote locations could be filtered by keywords, we used temporal information matching as an additional filtering approach.

*3.4.2.2 Temporal information matching.* Temporal information matching is the process of matching temporal expressions to date-time entities [41]. Temporal expressions can be given either explicitly (19 May 2021), implicitly (next Christmas) or relatively (on Tuesday). We extracted spaCy's date entities as temporal expressions. spaCy is able to extract date entities written in a variety of natural language formats, for example, 01-01-2019, Tomorrow, Valentine's day, Tuesday the 1$^{st}$ of Jan, last Monday etc.

To filter out tweets with remote location mentions, we formulated an assumption that a post with a temporal expression that does not coincide or overlap with the date the post was sent, will likely refer to a remote location. Examples of temporal expressions present in discarded tweets include; next week, yesterday, in two days, last Thursday etc.

## 3.5 Location extraction

A distinctive feature of the adopted spaCy pre-trained model is its split location entities. spaCy has location entities split into four classes. (1) Geopolitical Entities (GPE), (2) Facilities (FAC), (3) Organisations (ORG) and, (4) Locations (LOC). GPE entities represent administrative units such as countries, states and cities. FAC contains buildings, airports, highways and bridges. ORG includes companies, agencies and institutions. LOC defines street names, mountains, lakes and water bodies [36]. We extracted these four location entities separately.

Since spaCy extracts entities based on entity predictions [37], it is able to distinguish geographical from non-geographical entities, for instance, disambiguating the word 'Turkey' as either, a country, animal, food or a person's name based on sentence syntax. However, since most geocoding services do not take into account a tweet's context, geographical location ambiguities remain a challenge [42–44]. For example, Georgia could refer to Georgia a country in Europe or Georgia a state in the USA.

To reduce ambiguity when geocoding location entities, additional location information has to be attached to each entity. An ORG entity for instance "McDonald's" has to be combined with a GPE, FAC and or a LOC entity to the highest level of detail to achieve a "high probability" of being correctly geocoded. We use the term "high probability" to clarify the mere fact that more than one location entity, in this case "McDonald's", may as well be present in several cities, neighbourhoods or streets with similar geopolitical names.

To ensure high probability in the geocoded locations, we paired combinations of location entities within each post. The single location entities together with the combined location entities (Dual, Triple and Quad) resulted in fifteen location entity classes shown in Table 2.

## 3.6 Geocoding

Geocoding is a process of transforming human readable locations into latitude and longitude value pairs. We geocoded the extracted location entities using Nominatim and Google Maps geocoding API (Google Maps). Although several geocoders exist such as Esri Geocoding,

**Table 2. Location entity classes with corresponding examples.** The examples are given as extracted within the tweet text hence the presence of spelling errors and incorrect punctuation in some of the examples. The table is sorted by alphabetical order in each category.

| Category | | Location entity | Examples |
|---|---|---|---|
| Single location entities | 1 | *FAC* | Millennium Skate World |
| | 2 | *GPE* | Atlanta, Georgia |
| | 3 | *LOC* | Hyde Park |
| | 4 | *ORG* | Lincoln Trail College |
| Dual location entities | 5 | *FAC, GPE* | Millennium Skate World, Elmwood |
| | 6 | *FAC, LOC* | Revelation Beer Garden, Hudson Fields |
| | 7 | *LOC, GPE* | Hyde Park, Chicago, IL, USA |
| | 8 | *ORG, FAC* | LA Fitness, Indianapolis West |
| | 9 | *ORG, GPE* | Roundhouse Depot Brewing Co., Cleveland |
| | 10 | *ORG, LOC* | The Thayer Hotel, West Point |
| Triple location entities | 11 | *FAC, LOC, GPE* | Vizcaya Museumand Gardens, Biscayne Bay, miami |
| | 12 | *FAC, ORG, GPE* | Texas Familia; Liberty Plaza; annarbor |
| | 13 | *ORG, FAC, LOC* | Nam Knights Motorcycle Club, LLWF; Ground Zero; Lower Manhattan |
| | 14 | *ORG, LOC, GPE* | Courtyardby Marriott, Sumatra Mountain, Cranbury South Brunswick |
| Quad | 15 | *ORG, FAC, LOC, GPE* | Timbers; Providence Park; Cascadia; Vancouver |

GeoNames, Bing, HERE Maps, TomTom, etc. we restricted our analysis to Nominatim and Google Maps geocoders due to their worldwide popularity.

Nominatim uses crowdsourced data from OpenStreetMap (OSM) to match exact location names and return the associated coordinates for the input string [3]. Due to its crowdsourced nature, Nominatim contains place name aliases that allow for geocoding of often-used informal place names in tweets. However, a drawback of Nominatim is that it does not handle spelling errors well [15, 45]. For short and unstructured posts where spelling errors, noisy elements, and shorthand syntax are common, such as is the case with tweets, Nominatim fails to geocode a significant number of locations.

Contrasting to Nominatim, which returns locations based on exact name matches, Google Maps performs location cleaning and fuzzy matches on an input string and geocodes the resulting formatted address (see.Table 3). The advantage of this approach is that a higher number of locations are geocoded as spelling discrepancies and noisy elements are filtered out. However, the disadvantage is that some locations may be incorrectly formatted, which results in incorrect coordinates.

In the example input strings in Table 3, "**arizona, glendale, glendalearizona**" is correctly formatted to "**W Glendale Ave, Glendale, AZ 85301, USA**" despite the errors present in the input string. On the contrary '**chicagos**' was wrongly formatted and geocoded to an address in **Georgia** (GA), USA. Because each geocoder has its own pros and cons, we used both Nominatim and Google Maps API to geocode the fifteen categories of location entities defined in Table 2.

**Table 3. Examples of Google Maps geocoding API formatted and geocoded address from provided input string.**

| Input string | Formatted address | Judgement |
|---|---|---|
| arizona, glendale, glendalearizona | W Glendale Ave, Glendale, AZ 85301, USA | Correct |
| Chicagos | 4401 Shallowford Rd, Roswell, GA 30075, USA | Incorrect |

### 3.7 Displacement computations

As a validation, we computed the geodesic displacement between the inferred coordinates and the tweets' attached GNSS coordinate. For comparison with measurements from related studies [8, 9, 17, 35], we computed displacement values within 1 km, 10 km and 50 km radius values.

### 3.8 Generating statistical results

The displacement computations returned individual precision values for each location entity and displacement class. To compare performance of each dataset, we grouped location entities per category. We generated four mutually exclusive groups by combing corresponding location entities in each category. Our groups are defined as follows,

> *Group 1 = Quad location entities*
>
> *Group 2 = Triple location entities*
>
> *Group 3 = Dual location entities*
>
> *Group 4 = Single location entities*

Based on our justification of combining location entities (s. 3.5), we expect to obtain an inverse relationship between the precision and recall values, with Group 1 having a high precision and low recall and Group 4 having a low precision and high recall. We define our precision and recall evaluation matrix as follows:

$$\text{Precision} = \frac{\text{Number of geocoded locations within a k displacement value}}{\text{Total number of geocoded posts}}$$

$$\text{Recall} = \frac{\text{Number of successfully geocoded location entities}}{\text{Total number of extracted locations}}$$

## 4. Results

The aim of our paper was to enhance the geocoding precision of location inference models and investigate the influence of tweet source distribution when developing location inference models. In this section, we present the results of our proposed location inference model on Dataset A (proportionate sample of a geotagged dataset) and Dataset B (modelled tweet source distribution of a non-geotagged dataset).

Our first interaction with the data after tweet sample selection, was to pre-process the selected tweets (section 3.4). In Fig 3 we show the number of tweets that remained after each preprocessing step. From the figure it is apparent that Dataset A remained with a higher percentage of pre-processed tweets as compared to Dataset B. Dataset A remained with 94.8%

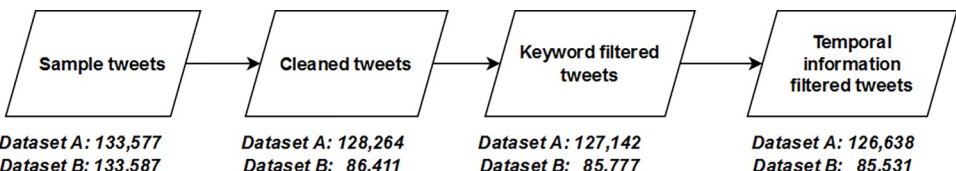

**Fig 3. Number of remaining tweets for each stage of the text preprocessing.** Sample tweets shows the original count of tweets in each dataset. Temporal information filtered tweets shows the remaining number of tweets after the last preprocessing step.

**Table 4. Mutually exclusive count of tweets with a location entity extracted in Dataset A and Dataset B.** The percentages of each location entity are computed as a fraction of the total tweets whilst the percentage of the total tweets is computed as a fraction of the remaining pre-processed posts, for each dataset. The table is sorted by descending order of number of extracted locations in Dataset A.

| Location Entity | Dataset A | | Dataset B | |
|---|---|---|---|---|
| GPE | 31,442 | (39.79%) | 6,373 | (35.92%) |
| ORG | 13,783 | (17.44%) | 5,654 | (31.87%) |
| FAC | 13,509 | (17.10%) | 1,688 | (9.52%) |
| ORG, GPE | 7,872 | (9.96%) | 1,651 | (9.31%) |
| FAC, GPE | 5,503 | (6.96%) | 909 | (5.12%) |
| ORG; FAC | 1,898 | (2.40%) | 298 | (1.68%) |
| LOC | 1,897 | (2.40%) | 691 | (3.90%) |
| LOC, GPE | 1,427 | (1.81%) | 149 | (0.84%) |
| FAC, ORG, GPE | 695 | (0.88%) | 145 | (0.82%) |
| FAC; LOC | 371 | (0.47%) | 54 | (0.30%) |
| ORG; LOC | 266 | (0.34%) | 60 | (0.34%) |
| FAC, LOC, GPE | 143 | (0.18%) | 28 | (0.16%) |
| ORG, LOC, GPE | 135 | (0.17%) | 24 | (0.14%) |
| ORG; FAC; LOC | 50 | (0.06%) | 13 | (0.07%) |
| ORG; FAC; LOC; GPE | 21 | (0.03%) | 3 | (0.02%) |
| **Total tweets** | **79,012** | **(62.39%)** | **17,740** | **(20.74%)** |

(126,638/133,577) pre-processed tweets, whilst, Dataset B remained with only 64% (85,531/133,587) tweets.

We ran our location extraction model on the pre-processed tweets for each dataset. In Table 4 we show the number and percentage of tweets where a corresponding location entity was extracted. Overall, Dataset A had a higher percentage of tweets with at least a single location entity extracted, 62.39%, as compared to Dataset B, 20.74%.

When we focus on individual location entities, we observe a general agreement on the ordering of tweets count per location entity. With an exception of the LOC entity, both datasets showed overall higher percentages of tweets with single location entities in comparison to combined location entities.

Following location extraction, we geocoded the extracted location entities using Nominatim and Goggle Maps (section 3.6). In Table 5 we show the percentage of geocoded posts for each location entity in each dataset. Primarily we note that higher percentages of location entities were geocoded for Dataset A compared to Dataset B for both geocoders. The difference in percentage of geocoded locations for the two datasets is more pronounced on Google Maps geocoded locations compared to Nominatim geocoded locations. With Google Maps, Dataset A had an average recall of 86.03% whilst the average recall was only 69.81% for Dataset B. With Nominatim, 69.42% of Dataset A locations were geocoded whilst 68.22% were geocoded for Dataset B.

Taking a closer look on Nominatim, we observe overall higher recall for single location entities compared to the combined location entities. With an exception of the FAC, LOC entity which had a recall of 70.24% and 55.88% for Dataset A and Dataset B, respectively, the recall for the combined location entities were below 50% for both datasets. The recall further reduced to less than 14.09% and less than 3.7%, for Dataset A and Dataset B, respectively, for triple and quad location entities.

After geocoding location entities, we computed the precision values within a 1 km, 10 km and 50 km radius of the tweets' GNSS position (section 3.7). In Table 6 we present the precision values of each location entity. Our first observation is that precision values are generally

**Table 5. Percentage of geocoded locations for each dataset when using either the Nominatim or Google Maps geocoder.** The table is arranged by alphabetical order of the location entity and category.

| Category | Location Entity | Nominatim | | Google Maps | |
|---|---|---|---|---|---|
| | | Dataset A | Dataset B | Dataset A | Dataset B |
| Single location entities | FAC | 68.31% | 63.13% | 81.41% | 65.49% |
| | GPE | 97.25% | 95.55% | 98.52% | 94.09% |
| | LOC | 87.58% | 79.16% | 85.32% | 72.80% |
| | ORG | 50.77% | 59.73% | 51.52% | 32.07% |
| Dual location entities | FAC, GPE | 48.91% | 43.13% | 97.99% | 96.68% |
| | FAC, LOC | 17.25% | 25.93% | 88.41% | 66.67% |
| | LOC, GPE | 70.24% | 55.88% | 97.91% | 95.10% |
| | ORG, FAC | 4.79% | 7.05% | 82.61% | 68.46% |
| | ORG, GPE | 31.24% | 30.39% | 93.85% | 88.92% |
| | ORG, LOC | 9.77% | 8.33% | 80.08% | 71.67% |
| Triple location entities | FAC, LOC, GPE | 14.09% | 3.23% | 97.32% | 93.55% |
| | FAC, ORG, GPE | 3.77% | 3.38% | 94.97% | 92.57% |
| | ORG, FAC, LOC | 0.00% | 0.00% | 86.00% | 76.92% |
| | ORG, LOC, GPE | 4.29% | 3.70% | 96.43% | 88.89% |
| Quad location entities | ORG, FAC, LOC, GPE | 0.00% | 0.00% | 95.24% | 100% |
| | **Total tweets geocoded** | **69.42%** | **68.22%** | **86.03%** | **69.81%** |

higher in Dataset A as compared to Dataset B, in both geocoders. Taking the Google Maps 50 km radius for example, whilst Dataset A reported 72.5% as the least precision value across all location entities, Dataset B reported 48.0% as the least precision value for the same geocoder and radius value.

We also observe differences in precision values between the two geocoders with Google Maps showing overall higher precision values compared to Nominatim. The differences

**Table 6. Precision values of each geocoded location entity geocoded with Nominatim for Dataset A and Dataset B.** The precision values are given for the 1 km, 10 km and 50 km displacement classes. Missing values indicate zero geocoded locations. The table is sorted alphabetically by location entities per each category.

| | Nominatim | | | | | | Google Maps | | | | | |
|---|---|---|---|---|---|---|---|---|---|---|---|---|
| | Dataset A | | | Dataset B | | | Dataset A | | | Dataset B | | |
| | 1 km | 10 km | 50km | 1km | 10km | 50km | 1km | 10km | 50km | 1km | 10km | 50km |
| FAC | 45.6 | 53.3 | 55.9 | 20.8 | 28.3 | 31.6 | 72.0 | 81.6 | 84.3 | 45.9 | 60.2 | 65.9 |
| GPE | 42.7 | 79.9 | 87.8 | 12.0 | 46.7 | 58.6 | 45.1 | 83.3 | 90.7 | 14.5 | 52.0 | 66.1 |
| LOC | 28.6 | 44.1 | 50.3 | 8.8 | 20.9 | 24.7 | 40.5 | 63.6 | 72.5 | 16.0 | 38.7 | 48.0 |
| ORG | 17.8 | 21.2 | 24.5 | 4.6 | 7.2 | 10.2 | 61.9 | 71.3 | 77.0 | 33.3 | 47.5 | 56.4 |
| FAC, GPE | 68.8 | 88.5 | 94.1 | 58.6 | 82.3 | 89.3 | 69.3 | 90.3 | 95.1 | 60.2 | 83.4 | 91.0 |
| FAC, LOC | 51.6 | 70.3 | 76.6 | 14.3 | 21.4 | 28.6 | 56.3 | 73.3 | 79.8 | 33.3 | 61.1 | 77.8 |
| LOC, GPE | 60.1 | 80.0 | 91.8 | 43.9 | 64.6 | 76.9 | 60.1 | 84.3 | 93.0 | 44.7 | 71.0 | 82.2 |
| ORG, FAC | 46.2 | 57.1 | 62.6 | 38.1 | 47.6 | 52.4 | 74.9 | 83.4 | 88.8 | 46.6 | 65.2 | 76.5 |
| ORG, GPE | 53.0 | 82.4 | 89.7 | 36.8 | 68.4 | 78.1 | 54.6 | 85.0 | 91.4 | 37.1 | 69.1 | 75.8 |
| ORG, LOC | 53.9 | 61.5 | 69.2 | 40.0 | 60.0 | 60.0 | 52.6 | 70.9 | 77.5 | 27.9 | 55.8 | 72.1 |
| FAC, LOC, GPE | 70.8 | 91.7 | 95.8 | 0.0 | 80.0 | 100 | 65.1 | 84.9 | 90.1 | 57.7 | 81.0 | 92.0 |
| FAC, ORG, GPE | 83.3 | 100 | 100 | 0.0 | 0.0 | 0.0 | 51.9 | 77.0 | 85.2 | 45.8 | 58.3 | 66.7 |
| ORG, FAC, LOC | - | - | - | - | - | - | 69.8 | 82.7 | 93.0 | 20.0 | 40.0 | 50.0 |
| ORG, LOC, GPE | 57.1 | 100 | 100 | 0.0 | 100 | 100 | 57.9 | 84.1 | 91.0 | 55.2 | 82.8 | 89.7 |
| ORG, FAC, LOC, GPE | - | - | - | - | - | - | 50.0 | 80.0 | 85.0 | 33.3 | 66.7 | 100 |

**Table 7. Precision and recall values per group for Datasets A and B derived from precision values returned by Nominatim and Google Maps.**

| | | Nominatim | | | | Google Maps | | | |
|---|---|---|---|---|---|---|---|---|---|
| | | Group 1 | Group 2 | Group 3 | Group 4 | Group 1 | Group 2 | Group 3 | Group 4 |
| **Dataset A** | 1 km | - | 75.5 | 60.7 | 39.2 | 50.0 | 55.4 | 61.9 | 53.1 |
| | 10 km | - | 96.8 | 84.0 | 65.1 | 80.0 | 79.3 | 86.1 | 80.6 |
| | 50 km | - | 98.4 | 91.3 | 71.3 | 85.0 | 87.0 | 92.1 | 86.8 |
| | *Recall* | *0* | *0.07* | *8.02* | *61.34* | *0.03* | *1.23* | *20.61* | *64.16* |
| **Dataset B** | 1 km | - | 0.0 | 45.5 | 10.4 | 33.3 | 47.1 | 45.3 | 21.9 |
| | 10 km | - | 25.7 | 72.3 | 31.6 | 66.7 | 63.1 | 73.1 | 51.4 |
| | 50 km | - | 28.6 | 81.0 | 39.6 | 100.0 | 71.8 | 81.0 | 63.2 |
| | *Recall* | *0* | *0.04* | *5.73* | *62.45* | *0.02* | *1.08* | *15.63* | *53.08* |

between the precision values of the two geocoders is more pronounced on the single location entities than on combined location entities. Taking the ORG entity as an example, whilst Nominatim geocoded less than 25% of the ORG entities within a 50km radius for both datasets, Google Maps geocoded more than 33% of the ORG entity within 1 km radius and ~56% within a 50 km radius.

If we take a closer look at the precision values, we observe higher precision values in combined location entities compared to single location entities. However, for the Nominatim geocoded locations, combinations with three or four location entities returned either 0% or 100% precisions.

We generated statistical results by grouping our location entities into four groups (section 3.8). In Table 7 we show the precision and recall values for Dataset A and Dataset B after geocoding with Nominatim and Google Maps, respectively. On overall, we observed higher precision and recall values in Dataset A as compared to Dataset B for both Nominatim and Google Maps.

In Dataset A, the highest precision values resulted from the Nominatim geocoder Group 2 (Triple location entities) combination. This combination resulted in **75.5%, 96.8%** and **98.4%** of tweets geocoded within a 1 km, 10 km and 50 km radius of the tweets' GNSS position. However the recall value, computed as a fraction of extracted locations, is only 0.07% for this combination. Dataset A, Google Maps, Group 4 (single location entities) resulted in the overall highest recall value of 64.16%. However the precision values for this combination reduced to **53.1%, 80.6%** and **86.8**% within a 1 km, 10 km and 50 km radius of the tweets' GNSS position, respectively. A somewhat balance between precision and recall value for Dataset A resulted from Google Maps Group 3 (dual location entities), where we achieved precision values of **61.9%, 86.1%** and **92.1%** within a 1 km, 10 km and 50 km radius respectively for a recall of 20.61%.

In Dataset B, the highest precision values tied between the Nominatim geocoder, Group 3 and the Google Maps geocoder, Group 3. Nominatim returned a slightly higher precision value in the 1 km radius (**45.5%**) compared to Google Maps (**45.3%**). However in the 10 km radius, Nominatim returned a lower precision value (**72.3%**) compared to Google Maps (**73.1%**). In the 50 km radius, both geocoders returned a precision value of **81.0%**. Considering the recall values, however, we observed that Nominatim returned lower recall (5.73%) compared to Google Maps (15.63%).

The highest overall recall in Dataset B (62.45%) resulted from Nominatim, Group 4. However the precision values for this combination was much lower with only **39.6%** of locations geocoded to within 50 km of the tweet's GNSS position. Google Maps Group 4 retuned higher precision values, **21.9%, 51.4%** and **63.2%** within a 1 km, 10 km and 50 km, respectively,

though from a slightly lower recall 53.08% compared to combination with the highest recall (Nominatim, Group 4).

## 5. Discussion

The objective of this paper was to enhance the geocoding precision for location inference of tweet text, and to investigate the influence of tweet source distribution when developing location inference models. In section 5.1, we provide interpretations of the obtained results. In section 5.2, we discuss the robustness and limitations of our developed methods.

### 5.1 Result discussion

We found that the **proportion of tweets from different sources** impacts the performance of the developed location inference model. Specifically, we found that we obtain higher model performance values when we design our location inference model on a dataset with a higher percentage of tweets emanating from third party applications compared to the native Twitter application.

Our Dataset A, which contained a proportionate sample of a geocoded dataset, returned overall higher precision and recall values compared to Dataset B, which contained a higher percentage of posts generated in native Twitter applications. Assuming similarities in text content between the tweet sources of geotagged and non-geotagged, this result implies that location inference models produce **higher performance values** than what would be obtained in a non-geotagged dataset. However, this assumption needs to be checked before location inference models are developed and adopted to a non-geotagged dataset.

Apart from data selection, we were able to confirm the well-known fact that the choice of the geocoder has an influence on the model performance. The model performance was overall higher for Google Maps geocoded locations as compared to Nominatim. The main drawback that we observed for Nominatim is that it struggles to resolve place names that have some mismatches within the OSM database which was evidenced by the low recall for combined location entities (s. Table 5). This observation goes in line with the findings of [15, 16, 45] who found the OSM Nominatim API to be unable to geocode a substantial percentage of location names.

Owing to Nominatim's struggle in resolving place names that did not have matches in the OSM database, the geocoder outperformed Google Maps precision values for combined location entities (s. Table 7). This finding can be attributed to Google Maps' tendency to format location mentions before geocoding which leads to an overall higher recall though at the cost of precision. What this result implies, is that in cases where precision is of more important than recall, it would be best to use Nominatim combined location entities than the Google Maps geocoder.

Albeit the differences in geocoder performances, most importantly, we found that our enhanced location inference model outweighs similar studies which have used sample tweets extracted from a geotagged dataset [8, 9, 17, 35]. In Table 8 we compare precision values returned from related studies with the precision values that resulted from our model combination with the highest precision, and model with a trade-off between precision and recall from Dataset A and Dataset B. With an exception of [9] who outweighed our precision values for Dataset B at a 50 km radius (83% vs 81%), our location inference approach outweighed the precision values for existing literature sources for both Dataset A (higher percentage of tweets generated from third party Twitter applications) and Dataset B (higher percentage of tweets generated from native Twitter applications).

**Table 8. Summary comparison of precision values between existing approaches which inferred a tweet's point of origin and our developed approaches.** The table is arranged by ascending order of precision value within each error distance.

| | | Method | Precision (%) |
|---|---|---|---|
| **DISTANCE (km)** | 1 km | Location keyword associations [17] | 18 |
| | | DBpedia staking approach [35] | 22 |
| | | Bayes model on foursquare [8] | 44.38 |
| | | *Dataset B Google Maps Group 3* | 45.3 |
| | | *Dataset B Nominatim Group 3* | 45.5 |
| | | *Dataset A Google Maps Group 3* | 61.9 |
| | | *Dataset A Nominatim Group 2* | 75.5 |
| | 10 km | DBpedia staking approach [35] | 37 |
| | | Location keyword associations [17] | 45 |
| | | Probability model of local words [46] | 56.7 |
| | | Ranking algorithm [9] | 60 |
| | | *Dataset B Nominatim Group 3* | 72.3 |
| | | *Dataset B Google Maps Group 3* | 73.1 |
| | | *Dataset A Google Maps Group 3* | 86.1 |
| | | *Dataset A Nominatim Group 2* | 96.8 |
| | 50 km | DBpedia staking approach [35] | 54 |
| | | Label propagation [47] | 65 |
| | | *Dataset B Nominatim Group 3* | 81.0 |
| | | *Dataset B Google Maps Group 3* | 81.0 |
| | | Ranking algorithm [9] | 83 |
| | | *Dataset A Google Maps Group 3* | 92.1 |
| | | *Dataset A Nominatim Group 2* | 98.4 |

## 5.2 Methods discussion

Considerably the most impactful decision for our analysis was the choice of the spaCy location extraction model. Using spaCy gave us **high** control over the extracted entities allowing us to test and combine precision values of various location combinations. The ability of spaCy to split location entities presents higher advantage over location extraction models with a single classification for location entities such as DBpedia, Geoparser.io, Stanford NLP, etc.

By combining geocoded tweets with a similar combination count of location entities (single location entities, dual location entities etc.), we produced four statistical groups from the 15 location entities. Whilst these four groups were sufficient for comparisons, a higher number of groups could have been formed by trying out various combinations of location entities. Whilst we were able to obtain high precision values from the four combinations, we acknowledge that prioritizing location entities based on precision values would have returned higher precision values.

Although our proposed method returned higher precision values compared to other studies, the lack of robustness in filtering out remote location mentions, impacted the performance of the location model. For instance, the post '*Happy birthday to my brother in Sydney*' contained neither keywords nor temporal information that could be used to filter out the remote location mention. For future work, we suggest to use **filtering approaches that take into account the grammatical structure** of the sentence in order to improve precision values.

The use of only tweets generated within a USA bounding box limits the **generalizability** of our location inference model on world-wide dataset. Since the USA has a high density of

mapped locations, it is possible that our high precision values were a result of bias towards the more popular USA cities. Since non-geocoded datasets are not constrained to specific countries or continents, there is a need to test the performance on our location inference on model on a world-wide dataset.

Another impacting factor that is worth mentioning is that users may also share **imprecise or contradicting** information about their location when sending the post. Since we compared the inferred locations with the tweets' GNSS position, the contradicting location mentions may have negatively affected the precision of the results. Contradicting location mentions are quite common in location sharing services where the user has the choice to tag a location of their choice such as Instagram.

## 6. Conclusion

This paper adopted existing location inference methods and presented an enhanced precision location inference method for inferring a tweet's point of origin using mentioned location entities within tweet text. Overall, we found that our proposed approach outweighs the performances of other approaches as described in the literature of previous research. Our model exhibits considerable higher precision values, which, in part, are resulting from an adopted spaCy model that allowed for high control over location entities.

Owing to the differences in tweet source distribution between a geotagged dataset, where location inference models are often trained on, and a non-geotagged dataset, where location inference models are designed for, we investigating the model performance on two datasets with different tweet source distributions. Our results confirm the well-known fact that the distribution of data from different tweet sources strongly influences the performance of location inference methods. Having a higher percentage of tweets generated from third applications showed an overall high performance as compared to having a higher percentage of tweets originating from native tweet applications. This finding thus question the applicability of location inference models on a non-geotagged dataset.

As part of our future work, we plan to investigate location mentions between geotagged and non-geotagged datasets at a tweet source level. By understanding location mentions between the two datasets, location inference methods can be modelled to suit more a non-geotagged dataset instead of a geotagged dataset. Additionally, owing to the geocoding limitations in both speed and cost, we plan to investigate multiple geocoders to carter for scaling issues.

## Supporting information

**S1 File. Extracting and geocoding locations in social media posts: A comparative analysis.**
(PDF)

**S1 Appendix.**
(DOCX)

## Author Contributions

**Conceptualization:** Helen Ngonidzashe Serere.

**Data curation:** Clemens Rudolf Havas.

**Funding acquisition:** Bernd Resch.

**Methodology:** Helen Ngonidzashe Serere.

**Supervision:** Bernd Resch.

**Validation:** Helen Ngonidzashe Serere.

**Writing – original draft:** Helen Ngonidzashe Serere.

**Writing – review & editing:** Helen Ngonidzashe Serere, Bernd Resch.

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
