## [Decision Letter · Decision Letter 0]

2 Feb 2022

PONE-D-21-27579High precision for location extraction for social media posts using spaCy, Nominatim and Google Maps API.PLOS ONE

Dear Dr. Serere,

Thank you for submitting your manuscript to PLOS ONE. After careful consideration, we feel that it has merit but does not fully meet PLOS ONE’s publication criteria as it currently stands. Therefore, we invite you to submit a revised version of the manuscript that fully addresses the points raised by the two reviewers. 

We look forward to receiving your revised manuscript.

Kind regards,

Zhenlong Li, Ph.D.

Academic Editor

PLOS ONE

https://journals.plos.org/plosone/s/file?id=ba62/PLOSOne_formatting_sample_title_authors_affiliations.pdf"

3. We note that [Figure 3] in your submission contain [map/satellite] images which may be copyrighted. All PLOS content is published under the Creative Commons Attribution License (CC BY 4.0), which means that the manuscript, images, and Supporting Information files will be freely available online, and any third party is permitted to access, download, copy, distribute, and use these materials in any way, even commercially, with proper attribution. For these reasons, we cannot publish previously copyrighted maps or satellite images created using proprietary data, such as Google software (Google Maps, Street View, and Earth). For more information, see our copyright guidelines: http://journals.plos.org/plosone/s/licenses-and-copyright.

a. You may seek permission from the original copyright holder of Figure 3 to publish the content specifically under the CC BY 4.0 license. 

4. Please include a copy of Table 4.

Reviewers' comments:

Reviewer's Responses to Questions

**Comments to the Author**

1. Is the manuscript technically sound, and do the data support the conclusions?

Reviewer #1: Partly

Reviewer #2: Yes

2. Has the statistical analysis been performed appropriately and rigorously? 

Reviewer #1: Yes

Reviewer #2: Yes

3. Have the authors made all data underlying the findings in their manuscript fully available?

Reviewer #1: No

Reviewer #2: Yes

4. Is the manuscript presented in an intelligible fashion and written in standard English?

Reviewer #1: Yes

Reviewer #2: Yes

5. Review Comments to the Author

Reviewer #1: The manuscript describes a methodology to geocode Twitter status messages’ point of origin, using the inferred location in the message textual content extracted using spaCy, and geocoded using a combination of Nominatim and Google Maps API. After the data collection, the workflow described consists on six distinct steps: 1) random sampling, 2) pre-processing, 3) extraction of location entities, 4) geocoding, 5) estimation of the displacement compared to the GNSS position “ground truth”, and 6) statistical analysis of the results within 3 radius thresholds.

The topic is current and can be of interest for the readers. The text is well structured and the references are adequate. The methodology is correctly described, and the results are presented adequately. The major limitations are also explained in the discussion. The conclusion would require more work as it reads more like an extended version of the abstract, although some paragraphs in section 5 could be considered part of the conclusion.

I have included the comments divided in three sections, 1) major issues, 2) minor issues, and 3) formatting/spelling errors.

MAJOR ISSUES

I think the use of “high precision” in title is misleading, because it is subjective as “high” precision/accuracy has different meanings depending on the context. I would suggest “(improved/enhanced) precision” as the method discussed in the manuscript claims better accuracy than the state of the art used as benchmark. Another aspect in the title that could be made more precise is substituting “social media” by “twitter” as no other social media service was analysed.

The explicit georeferenced data used as the “ground truth” in the accuracy assessment is oftentimes not accurate. Beyond limitations in the location accuracy of the mobile devices themselves (especially in dense urban areas) for automatically geotagged social media posts, some metadata consists in approximate coordinates (e.g. the centroid of an administrative division), or relies on unreliable user input. I think that it is necessary that the manuscript discusses these limitations as they have the potential to significantly alter the results.

The reason for the random selection mentioned on paragraph 182-184 is explained as being chosen for “computational reasons”. I think that the reason because it was not feasible to process the complete set of 5.6 million tweets should be clarified, in particular whether it was impossible or it was a practical decision, as well as the justification of the chosen sample size.

In addition to the decision to work with a subset of the data, I think that is important to justify why was not stratified sampling chosen instead, weighted on the proportion of messages from each source regardless of whether they had location data or not. The explained approach makes Instagram sources overrepresented, possibly leading to overfit, which can severely impact the performance if the whole dataset was considered, which is the stated objective of the authors. In addition, another issue that should be considered is that the demographics of Instagram and Twitter are not aligned and therefore one may not be representative of another.

The minor issues identified follow in the next section. However, I would like to point out that while they are minor issues, they are chained from one to the next in a long process of manual cleaning and feature engineering steps, and their combination could lead to problems with overfitting and make the results challenging to replicate, especially since the described approach does not include separate training and test sets. This should be investigated further and mentioned in the discussion.

MINOR ISSUES

I would suggest that keywords would not repeat worlds already in the title (spaCy, Nominatim, Google Maps API).

In lines 13-14 (“With a fraction of less than 10% geotagged posts, over 90% of posts are not analysed”) and lines 35-36 (“This means that more than 90% of geosocial media posts cannot be used for analysis”) the authors should clarify that it refers to spatial analysis only, as other content (semantics, multimedia content, network connections…) can be analysed without location data. The percentage is aligned with the results of this reviewer’s findings, but should include a reference as not all readers will be familiar with this imbalance.

The research uses the Nominatim and Google geocoders, but I think an explanation of whether other geocoders were also considered and why they were discarded should be included.

The order in which the steps described in subsection 3.2 are conducted can affect the outcome. In paragraph 165-173, it would be interesting to see a number of the distribution of tweets per source, sorted in descending order. Alternatively, figure 3 (left) could be presented as a chart. In figure 2, the number of remaining tweets after each step should be included, or the number of discarded ones, perhaps graphically. It would also be interesting to know the 2 additional discarded sources in paragraph 174-181.

For easier understanding, figure 3 should be split into two, and the map placed in a separate figure as both are very loosely related. If they are kept together, the caption should refer to both.

The number or proportion of messages translated (line 197, lines 406-410) should be stated. Likewise, in the case of messages removed by keyword filtering and temporal information matching (section 3.4.2), the resulting number of samples after both processes are explained in line 240, but not individually (keyword, temporal). The numbers could also be included in figure 5.

I am not familiar enough with Nominatim, but the Google geocoder often returns multiple locations to a query, and I assume that the first one was the only considered (line 441). However, the text mentions “most probable”, which should be clarified (is it the most precise? e.g. “ROOFTOP”). It is also possible that Google results are biased towards returning SOME result (for commercial reasons) and therefore provide a result even for very ambiguous queries (lines 435-438). In section 3.6 it would also be interesting to measure whether it exists a consensus in both geocoders’ results.

From the text it is not clear the grouping explanation in section 3.8, I think it could be easier to understand with an additional column with the corresponding grouping letter(s) in table 1. I think it would also be necessary to provide an explanation on why were not tested individually, or on the contrary, reduce the number of groups.

Figure 6 should be sorted according to the number of entities extracted, and table 1 could follow this order for easier comparison. For clarity it would also be possible to sort only within the 1/2/3 entity groups separately. Although charts with dual axes (in this case vertical) are not advisable in general, it seems to work fine in this case to summarize the results, although the data corresponding to the second axis could appear above or below in a subpanel with its own vertical scale, sharing the horizontal scale. In addition, I think that the use of a line chart instead of points/bars suggest a relationship which is not suitable for the data at hand, and reads like a time series.

TYPOS (not exhaustive)

References

• Consolidate multiple references, e.g. [7], [8] and [9] to [7, 8, 9] in line 52, or place after each item in lines 53-54 if applicable

• Extra [10] reference at the end of line 55

• “Orphan” references without subject, such as [11] in line 59, [14] in line 71, [17] in line 85, [18] in line 89, [6] in line 96… (in APA style it would be acceptable)

• Unnamed references, e.g. “done by [12]” in line 63, [13] in line 65…

Missing references for some claims

• Number of tweets with location data (line 34-35)

• “good performance” of spaCy on twitter data (line 132)

Use of informal language

• “we see” in line 38

• “tremendous” in line 52

• “64+” language models in line 128

• “other such” in line 129

• “cells” in line 203 is Pandas-specific

Formatting

• The bold text in lines 55, 70 and 85 would seem to work better as subsections of section 2

• Reference not found in line 168

• “Appendix” in line 222

• Reference not found in lines 403-404

Misspellings

• “profiled” in line 59

• Lower case Google in line 61

• Lower case python in line 127

• “\\” in line 156

Reviewer #2: The study aims to geocode the non-geotagged social media content with a proposed methodology. I believe the study should include the geo-parsing studies (geoparsepy, geoparser.io, unlock etc.) in the introduction part to provide wide perspectives of how spaCy works.

In the geocoding part the proposed 4 approach should be exemplified step by step. How the Nominatim + Google Maps API and Google Maps API on Nominatim geocoded locations impact the results?

For the displacement granularity (1, 10, 50km), it might be good to look at the city or county area and re-determine the granularity.

In the methodology part, how the data flow tackles with the duplicating place names (chain stores, coffee shops, county names) should be identified. A flow diagram would be more readable for it.

Used Twitter data are generated from several platforms (Instagram, foursquare,..) and this means the attached location copied their places libraries and easily be manipulated manually. This might be good point to add this while discussing the locational accuracy of each tweet.

6. PLOS authors have the option to publish the peer review history of their article (what does this mean?). If published, this will include your full peer review and any attached files.

Reviewer #1: No

Reviewer #2: **Yes: **Ayse Giz Gulnerman

---

## [Author Response · Author response to Decision Letter 0]

17 Nov 2022

Rebuttal letter 

PONE-D-21-27579

High precision for location extraction for social media posts using spaCy, Nominatim and Google Maps API.

PLOS ONE

Thank you for the opportunity to address the comments from the Reviewers and the Committee. The authors hope that the Reviewers and Editors will be satisfied with the amendments which we have made to the manuscript after taking into account the feedback provided.

**Report from PLOS ONE**

Thank you for submitting your manuscript to PLOS ONE. After careful consideration, we feel that it has merit but does not fully meet PLOS ONE’s publication criteria as it currently stands. Therefore, we invite you to submit a revised version of the manuscript

that fully addresses the points raised by the two reviewers. 

https://journals.plos.org/plosone/s/file?id=ba62/PLOSOne_formatting_sample_title_authors_affiliations.pdf"

Response: The authors have ensured that the manuscript meets the PLOS ONE’s style requirements. 

Response: The authors will make all data available upon acceptance. 

3. We note that [Figure 3] in your submission contain [map/satellite] images which may be copyrighted. All PLOS content is published under the Creative Commons Attribution License (CC BY 4.0), which means that the manuscript, images, and Supporting Information files will be freely available online, and any third party is permitted to access, download, copy, distribute, and use these materials in any way, even commercially, with proper attribution. For these reasons, we cannot publish previously copyrighted maps or satellite images created using proprietary data, such as Google software (Google Maps, Street View, and Earth). For more information, see our copyright guidelines: http://journals.plos.org/plosone/s/licenses-and-copyright.

a. You may seek permission from the original copyright holder of Figure 3 to publish the content specifically under the CC BY 4.0 license. 

Response: The authors have removed [Figure 3] and supplied a replacement Figure that complies with the CC BY 4.0 licence. The caption has also been modified in line with the new figure. 

4. Please include a copy of Table 4.

Response: The authors have made sure to separate tables and figures likewise.

Reviewers' comments:

Reviewer's Responses to Questions

Comments to the Author

1. Is the manuscript technically sound, and do the data support the conclusions?

Reviewer #1: Partly

Reviewer #2: Yes

2. Has the statistical analysis been performed appropriately and rigorously? 

Reviewer #1: Yes

Reviewer #2: Yes

3. Have the authors made all data underlying the findings in their manuscript fully available?

Reviewer #1: No

Reviewer #2: Yes

4. Is the manuscript presented in an intelligible fashion and written in standard English?

Reviewer #1: Yes

Reviewer #2: Yes

5. Review Comments to the Author

Reviewer #1: The manuscript describes a methodology to geocode Twitter status messages’ point of origin, using the inferred location in the message textual content extracted using spaCy, and geocoded using a combination of Nominatim and Google Maps API. After the data collection, the workflow described consists on six distinct steps: 1) random sampling, 2) pre-processing, 3) extraction of location entities, 4) geocoding, 5) estimation of the displacement compared to the GNSS position “ground truth”, and 6) statistical analysis of the results within 3 radius thresholds.

The topic is current and can be of interest for the readers. The text is well structured and the references are adequate. The methodology is correctly described, and the results are presented adequately. The major limitations are also explained in the discussion. The conclusion would require more work as it reads more like an extended version of the abstract, although some paragraphs in section 5 could be considered part of the conclusion.

I have included the comments divided in three sections, 1) major issues, 2) minor issues, and 3) formatting/spelling errors.

Response: The authors thank the Reviewer for the positive feedback. We have addressed the raised points as outlined in the paragraphs below. 

MAJOR ISSUES

I think the use of “high precision” in title is misleading, because it is subjective as “high” precision/accuracy has different meanings depending on the context. I would suggest “(improved/enhanced) precision” as the method discussed in the manuscript claims better accuracy than the state of the art used as benchmark. Another aspect in the title that could be made more precise is substituting “social media” by “twitter” as no other social media service was analysed.

Response: The authors thank the Reviewer for pointing the discrepancy between the paper and the title. Following other major issues outlined below, the authors have changed the title of the paper to: 

“Influence of data selection on Twitter location inference models”

The explicit georeferenced data used as the “ground truth” in the accuracy assessment is oftentimes not accurate. Beyond limitations in the location accuracy of the mobile devices themselves (especially in dense urban areas) for automatically geotagged social media posts, some metadata consists in approximate coordinates (e.g. the centroid of an administrative division), or relies on unreliable user input. I think that it is necessary that the manuscript discusses these limitations as they have the potential to significantly alter the results.

Response: Thank you for your observation. The authors agree that georeferenced tweet data is often times not accurate and may be given at higher scales leading to location estimates and not the actual position from which a post originated. However, the authors would like to point out that the study used only tweets with a precise GNSS position (Latitude and Longitude coordinate pairs) and not a place tag. Thus the accuracy assessment was based on the most precise data. The authors have adjusted the explanation in the manuscript to provide more clarification of the accuracy of the ground truth data, taking into account the raised concern. 

The reason for the random selection mentioned on paragraph 182-184 is explained as being chosen for “computational reasons”. I think that the reason because it was not feasible to process the complete set of 5.6 million tweets should be clarified, in particular whether it was impossible or it was a practical decision, as well as the justification of the chosen sample size.

Response: Thank you for identifying this lack of clarity. The authors revised the method section including the data selection procedure and have provided clarity to the data selection.

In addition to the decision to work with a subset of the data, I think that is important to justify why was not stratified sampling chosen instead, weighted on the proportion of messages from each source regardless of whether they had location data or not. The explained approach makes Instagram sources overrepresented, possibly leading to overfit, which can severely impact the performance if the whole dataset was considered, which is the stated objective of the authors. In addition, another issue that should be considered is that the demographics of Instagram and Twitter are not aligned and therefore one may not be representative of another.

Response: The authors thank the reviewer for pointing this out as it seems to be a very important overlooked point. Due to this comment, the authors have revised the complete paper and have looked at the influences of having different data sources within the training dataset. More specifically, the authors checked the performance of the developed location inference method on a proportionate sample of a geocoded dataset (having a higher percentage of Instagram tweets) and when a representation of non-geocoded dataset is used, that is using a non-geocoded representation of Twitter data sources. From their analysis, the authors have found that indeed choice of data selection used has high influence on the performance of a location inference models. 

The minor issues identified follow in the next section. However, I would like to point out that while they are minor issues, they are chained from one to the next in a long process of manual cleaning and feature engineering steps, and their combination could lead to problems with overfitting and make the results challenging to replicate, especially since the described approach does not include separate training and test sets. This should be investigated further and mentioned in the discussion.

Response: Thank you for your pointing this out. The authors have used two separate datasets. The first dataset consists of a proportionate sample of twitter source strata from a geocoded dataset. The second dataset consists of a strategically selected dataset that models the twitter source ratios in a non-geocoded dataset. 

MINOR ISSUES

I would suggest that keywords would not repeat worlds already in the title (spaCy, Nominatim, Google Maps API).

Response: The authors agree on the unnecessary repetition between the keywords and the title. The authors have made sure that the keywords used do not repeat with words in the new title. 

In lines 13-14 (“With a fraction of less than 10% geotagged posts, over 90% of posts are not analysed”) and lines 35-36 (“This means that more than 90% of geosocial media posts cannot be used for analysis”) the authors should clarify that it refers to spatial analysis only, as other content (semantics, multimedia content, network connections…) can be analysed without location data. The percentage is aligned with the results of this reviewer’s findings, but should include a reference as not all readers will be familiar with this imbalance.

Response: The authors agree and have explicitly stated that the limitation is only with spatial analysis. The authors have inserted the citations that go along with the given statement. 

The research uses the Nominatim and Google geocoders, but I think an explanation of whether other geocoders were also considered and why they were discarded should be included.

Response: Thank you for pointing this out. The authors have provided a reason to the use of only these two geocoders. 

The order in which the steps described in subsection 3.2 are conducted can affect the outcome. In paragraph 165-173, it would be interesting to see a number of the distribution of tweets per source, sorted in descending order. 

Response: The authors have added Table 1 which shows the number of Tweets per source sorted by descending order of the total tweets and the corresponding number of selected posts per source for the two datasets used.

Alternatively, figure 3 (left) could be presented as a chart.

Response: The mentioned Figure 3 (left) has been added separately as Table 1.

In figure 2, the number of remaining tweets after each step should be included, or the number of discarded ones, perhaps graphically.

Response: The authors have revised Figure 2 and added the number of discarded tweets, remaining posts and the remaining sources for each stage. 

It would also be interesting to know the 2 additional discarded sources in paragraph 174-181.

Response: Thank for pointing this out. The authors have mentioned the two additional discarded sources in lines 185-186 and have added justification for discarding the sources between lines 181-183.

For easier understanding, figure 3 should be split into two, and the map placed in a separate figure as both are very loosely related. If they are kept together, the caption should refer to both.

Response: The authors agree with your observation and have split figure 3 (left) to Table 1 showing the distribution of tweets per post and Figure 3 (right) to Figure 3 showing the distribution of the geotagged posts within the bounding box. 

The number or proportion of messages translated (line 197, lines 406-410) should be stated.

Response: Thank you. The authors agree that the proportion of translated tweets need to be specified. For the new analysis however, the authors decided to work with only English tweets hence no translation was done.

Likewise, in the case of messages removed by keyword filtering and temporal information matching (section 3.4.2), the resulting number of samples after both processes are explained in line 240, but not individually (keyword, temporal). The numbers could also be included in figure 5.

I am not familiar enough with Nominatim, but the Google geocoder often returns multiple locations to a query, and I assume that the first one was the only considered (line 441). However, the text mentions “most probable”, which should be clarified (is it the most precise? e.g. “ROOFTOP”).

Response: Indeed Google Maps returns multiple locations to a single query. The authors had used the term most probable to indicate the first location returned by Google Maps as the documentation states that Google Maps returns the most popular location as the first result. However, the authors acknowledge the ambiguity in the statement and have dropped the phrase most probable.

It is also possible that Google results are biased towards returning SOME result (for commercial reasons) and therefore provide a result even for very ambiguous queries (lines 435-438).

In section 3.6 it would also be interesting to measure whether it exists a consensus in both geocoders’ results.

Response: Thank you. The authors agree with the raised point however the authors could not include the results of the consensus as the manuscript was getting rather large.

From the text it is not clear the grouping explanation in section 3.8, I think it could be easier to understand with an additional column with the corresponding grouping letter(s) in table 1.

Response: Thank you for the comment. The authors have added elaborative texts and equations to clarify the groupings used in section 3.8.

I think it would also be necessary to provide an explanation on why were not tested individually, or on the contrary, reduce the number of groups.

Response: Thank you. The authors have added explanations to clarify the decision for the tested location entities. 

Figure 6 should be sorted according to the number of entities extracted, and table 1 could follow this order for easier comparison. For clarity it would also be possible to sort only within the 1/2/3 entity groups separately. 

Response: Thank you for the comment. The authors have sorted figure 6 and Table 1 accordingly. 

Although charts with dual axes (in this case vertical) are not advisable in general, it seems to work fine in this case to summarize the results, although the data corresponding to the second axis could appear above or below in a subpanel with its own vertical scale, sharing the horizontal scale.

Response: Thank you for the comment. We have replaced the figure completely.

In addition, I think that the use of a line chart instead of points/bars suggest a relationship which is not suitable for the data at hand, and reads like a time series.

Response: The authors agree with the reviewer’s observation. The line chart has been removed and the values previously represented as a line chart have been tabulated under table 3. 

TYPOS (not exhaustive)

References

• Consolidate multiple references, e.g. [7], [8] and [9] to [7, 8, 9] in line 52, or place after each item in lines 53-54 if applicable.

Response: The references have been consolidated. 

• Extra [10] reference at the end of line 55

Response: Thank you. This has been corrected.

• “Orphan” references without subject, such as [11] in line 59, [14] in line 71, [17] in line 85, [18] in line 89, [6] in line 96… (in APA style it would be acceptable)

Response: Thank you for pointing this out. The authors have adjusted the text to remove Orphan references.

• Unnamed references, e.g. “done by [12]” in line 63, [13] in line 65…

Response: Thank you. The authors are not clear what the unnamed references are referring to in this part. If you refer to the insertion of the reference at the end of the sentence, the authors have tried to include references in text, however for some sentences have been constructed to include the reference at the end of the line.

Missing references for some claims

• Number of tweets with location data (line 34-35)

Response: Thank you. The authors have included all references which were missing.

• “good performance” of spaCy on twitter data (line 132)

Response: Thank you. The authors have corrected this.

Use of informal language

• “we see” in line 38

• “tremendous” in line 52

• “64+” language models in line 128

• “other such” in line 129

• “cells” in line 203 is Pandas-specific

Response: Thank you. The authors have corrected and revised the manuscript to reduce informal language.

Formatting

• The bold text in lines 55, 70 and 85 would seem to work better as subsections of section 2

Response: Thank you. The authors have corrected this.

• Reference not found in line 168

Response: Thank you. The authors have corrected this.

• “Appendix” in line 222

Response: The authors thank the reviewer. However, the authors have failed to identify the error in the outlined word.

• Reference not found in lines 403-404

Response: Thank you. The authors have corrected this.

Misspellings

• “profiled” in line 59

• Lower case Google in line 61

• Lower case python in line 127

• “\\” in line 156

Response: The authors have addressed the comments above and have revised the manuscript in check for additional misspellings. 

Reviewer #2: The study aims to geocode the non-geotagged social media content with a proposed methodology. I believe the study should include the geo-parsing studies (geoparsepy, geoparser.io, unlock etc.) in the introduction part to provide wide perspectives of how spaCy works.

Response: The authors thank the Reviewer for the positive feedback and have addressed the raised points as outlined in the paragraphs below. 

In the geocoding part the proposed 4 approach should be exemplified step by step. How the Nominatim + Google Maps API and Google Maps API on Nominatim geocoded locations impact the results?

Response: Thank you for the response. The authors have reduced the approaches to only Nominatim and Google Maps API in line with the new analysis made.

For the displacement granularity (1, 10, 50km), it might be good to look at the city or county area and re-determine the granularity.

Response: The authors agree that the granularities could also be defined at city or country level. However, the objective of the study was to geolocate with high precision the position of the user at the time of sending each post. For this reason, the authors felt that the displacement of up to 50km was adequate. 

In the methodology part, how the data flow tackles with the duplicating place names (chain stores, coffee shops, county names) should be identified. A flow diagram would be more readable for it.

Response: Thank you. Unfortunately the authors do not understand what is meant by how the data flow tackles with duplicating place names. 

Used Twitter data are generated from several platforms (Instagram, foursquare,..) and this means the attached location copied their places libraries and easily be manipulated manually. This might be good point to add this while discussing the locational accuracy of each tweet.

Response: Thank you for the response. The authors have added a new analysis and investigated the different data sources of Twitter data as such the authors have provided a discussion of the location accuracies of the returned tweets.

6. PLOS authors have the option to publish the peer review history of their article (what does this mean?). If published, this will include your full peer review and any attached files.

Do you want your identity to be public for this peer review? For information about this choice, including consent withdrawal, please see our Privacy Policy.

Reviewer #1: No

Reviewer #2: Yes: Ayse Giz Gulnerman

Response: The above listed items have been addressed in the revised manuscript. The stated references within the manuscript are shown under the Reference list.

Reference list

[1] Huang B, Carley KM. A large-scale empirical study of geotagging behavior on twitter. Proc 2019 IEEE/ACM Int Conf Adv Soc Networks Anal Mining, ASONAM 2019 2019:365–73. https://doi.org/10.1145/3341161.3342870.

---

## [Decision Letter · Decision Letter 1]

26 Dec 2022

PONE-D-21-27579R1Enhanced geocoding precision for location inference of tweet text using spaCy, Nominatim and Google Maps. A comparative analysis of the influence of data selection.PLOS ONE

Dear Dr. Serere,

Thank you for submitting your manuscript to PLOS ONE. After careful consideration, we feel that it has merit but does not fully meet PLOS ONE’s publication criteria as it currently stands. Therefore, we invite you to submit a revised version of the manuscript that addresses the points raised during the review process.

In particular, while two referees propose minor changes, the third had made some criticisms.

We look forward to receiving your revised manuscript.

Kind regards,

Pierluigi Vellucci

Academic Editor

PLOS ONE

Journal Requirements:

Reviewers' comments:

Reviewer's Responses to Questions

**Comments to the Author**

1. If the authors have adequately addressed your comments raised in a previous round of review and you feel that this manuscript is now acceptable for publication, you may indicate that here to bypass the “Comments to the Author” section, enter your conflict of interest statement in the “Confidential to Editor” section, and submit your "Accept" recommendation.

Reviewer #1: (No Response)

Reviewer #3: All comments have been addressed

Reviewer #4: All comments have been addressed

2. Is the manuscript technically sound, and do the data support the conclusions?

Reviewer #1: Yes

Reviewer #3: Yes

Reviewer #4: Partly

3. Has the statistical analysis been performed appropriately and rigorously? 

Reviewer #1: Yes

Reviewer #3: Yes

Reviewer #4: Yes

4. Have the authors made all data underlying the findings in their manuscript fully available?

Reviewer #1: Yes

Reviewer #3: Yes

Reviewer #4: No

5. Is the manuscript presented in an intelligible fashion and written in standard English?

Reviewer #1: Yes

Reviewer #3: Yes

Reviewer #4: Yes

6. Review Comments to the Author

Reviewer #1: The authors have adequately addressed the comments mentioned in the first review round and the manuscript and this reviewer feels that it is now acceptable for publication after correcting the minor issues listed below. In addition, the paper has been revised to include the analysis of the influence of having different data sources, for example Instagram, and have included a discussion in section 5.1. In the case of Instagram, the reviewer would like to point out that the location is oftentimes introduced by the users themselves manually in a map, instead of using GNSS, which could arguably explain the differences in performance.

There are some identified minor issues (not exhaustive)

- GNSS should be defined elsewhere than the abstract on first mention.

- "Error! Reference source not found" in lines 169, 178, 179, 180, 181, 192, 212, 222, 257, 290, 308, 312, 316, 364, 380, 446

- Decimal comma instead of thousands separator in line 347

- Table 7 is shown almost a full page after being referred in line 398

Duplicated section 5.1 in line 466 that should be 5.2

As a final suggestion, it would be interesting to know if the recent changes in Twitter ownership could impact the applicability of the findings in the future.

Reviewer #3: The paper presents a method to geolocate tweets based on their textual content. Actually, location inference is at the core of the subject, aiming to get as much geocoding precision as possible from references to places made in the (usually terse) tweet text. Validation is achieved using a fully geotagged tweet dataset, in which geographic coordinates are associated to each message.

Being this a first review on a second-version paper, I did not find much cause for major revisions. Methodologically the paper is sound, used a good ground-truth-based validation process, and presents its findings accurately and in sufficient detail. Data are available online at Harvard’s repository.

The paper’s subject has been intensively explored in years past, especially in the early 2010s, a time in which most people did not care or were unaware of privacy concerns and allowed Twitter to release the geolocation at which a tweet was issued. This provided researchers with lots of geotagged messages to explore, and therefore several interesting and important articles can be found from around that time. However, this set of contributions is left out of the literature review, which concentrates on more recent work. It’s a pity that there has been no connection with works from a time in which Twitter provided much more ground truth data than today, when users in general have blocked their location to be associated with tweets.

There are even interesting works from 10 or 15 years ago, or even earlier, that revolve around the so-called “spatial prepositions”, expressions that are used in text in relation to toponyms and other references to places. In a sense, this is what the present paper proposes, including the comparison of toponym candidates against a reference dataset (some gazetteer back then; OSM, spaCy and Google now). The idea of single, double, triple references in a single message has also been used successfully in the past, trying to geolocate traffic events from Twitter.

Nevertheless, at the time of these contributions, data was not usually provided for replicability purposes, so the authors could only have access to ideas and techniques that could be compared to their proposals.

In conclusion, the paper is sound, is very well written and organized, uses a good and publicly available dataset with adequate annotations with which to perform analyses and comparisons. The observation in the conclusions that the geocoder has a significant influence in the results comes as no surprise, since this has been reported in several of the works I mentioned earlier. It also must be said that some of the ideas and techniques that are employed in the paper have been tried before, including (1) extraction of location indications from the tweet content, (2) analysis of expressions that indicate a location (spatial prepositions), (3) use of gazetteers and addressing databases (not necessarily embedded in a geocoding product), (4) improved reliability of messages in which multiple references appear, (5) using past messages and user relationships in the definition of a tweet's location. I’d only wish that the authors’ literature review reached further back so that they could get to know and recognize some ideas presented earlier in geocoding, geographic information retrieval, toponym resolution, location-based social networks and other subfields.

Reviewer #4: - The major concern is that the contributions of this article are not clear. The authors seem to use existing data, existing methods, and existing tools.

- At the same time the authors claim that they have developed a "new high precision geocoding approach" - The authors should explicitly describe what their approach is and how it is different to existing approaches.

- The motivation for filtering based on apps is not clear. Sure, sampling based on data source will impact the result because different apps create different user behavior and data. But why is this relevant? And why is this relevant for this paper in which a new geocoding approach is being developed?

7. PLOS authors have the option to publish the peer review history of their article (what does this mean?). If published, this will include your full peer review and any attached files.

Reviewer #1: No

Reviewer #3: No

Reviewer #4: No

---

## [Author Response · Author response to Decision Letter 1]

9 Feb 2023

Rebuttal letter

Response: The authors would like to thank the Academic editors and the Reviewers for the opportunity to address the comments. The authors hope that the Editors and Reviewers will be satisfied with the amendments which we have made to the manuscript after taking into account the feedback provided.

Reviewer #1: The authors have adequately addressed the comments mentioned in the first review round and the manuscript and this reviewer feels that it is now acceptable for publication after correcting the minor issues listed below. In addition, the paper has been revised to include the analysis of the influence of having different data sources, for example Instagram, and have included a discussion in section 5.1. In the case of Instagram, the reviewer would like to point out that the location is oftentimes introduced by the users themselves manually in a map, instead of using GNSS, which could arguably explain the differences in performance.

Response: The authors thank the reviewer for the positive feedback. The authors agree with the reviewer that Instagram does not always rely on GNSS for location sharing. The last paragraph of the discussion section points out to the reviewers point.

There are some identified minor issues (not exhaustive)

- GNSS should be defined elsewhere than the abstract on first mention.

Response: Thank you: The authors have removed the GNSS from the abstract and defined it on line 148 which is its first appearance in the text. 

- "Error! Reference source not found" in lines 169, 178, 179, 180, 181, 192, 212, 222, 257, 290, 308, 312, 316, 364, 380, 446

Response: The authors thank the reviewer for pointing this out. The references must have been altered during saving of the document. The authors have corrected all instances where cross reference is not found and hope that the cross reference will not have additional issues.

- Decimal comma instead of thousands separator in line 347

Response: Thank you. The authors have fixed the document and cross-checked for any other similar instances. 

- Table 7 is shown almost a full page after being referred in line 398

Response: The authors thank the reviewer for pointing this out and have moved Table directly below its first mention.

Duplicated section 5.1 in line 466 that should be 5.2

Response: Thank you. The authors have corrected the duplicate.

As a final suggestion, it would be interesting to know if the recent changes in Twitter ownership could impact the applicability of the findings in the future.

Response: The authors thank the reviewer for bringing this up. Yes, indeed there has been a lot of speculation on foreseeable impacts of most Twitter analysis. In terms of future applicability of location inference models, the authors do not foresee any major impact. In a probably worst case scenario where real time tweets are not publicly available or a huge number of people uninstall Twitter, location inference models can always be applied to historical dataset or adopted to match different social media datasets.

Reviewer #3: The paper presents a method to geolocate tweets based on their textual content. Actually, location inference is at the core of the subject, aiming to get as much geocoding precision as possible from references to places made in the (usually terse) tweet text. Validation is achieved using a fully geotagged tweet dataset, in which geographic coordinates are associated to each message.

Being this a first review on a second-version paper, I did not find much cause for major revisions. Methodologically the paper is sound, used a good ground-truth-based validation process, and presents its findings accurately and in sufficient detail. Data are available online at Harvard’s repository.

The paper’s subject has been intensively explored in years past, especially in the early 2010s, a time in which most people did not care or were unaware of privacy concerns and allowed Twitter to release the geolocation at which a tweet was issued. This provided researchers with lots of geotagged messages to explore, and therefore several interesting and important articles can be found from around that time. However, this set of contributions is left out of the literature review, which concentrates on more recent work. It’s a pity that there has been no connection with works from a time in which Twitter provided much more ground truth data than today, when users in general have blocked their location to be associated with tweets.

There are even interesting works from 10 or 15 years ago, or even earlier, that revolve around the so-called “spatial prepositions”, expressions that are used in text in relation to toponyms and other references to places. In a sense, this is what the present paper proposes, including the comparison of toponym candidates against a reference dataset (some gazetteer back then; OSM, spaCy and Google now). The idea of single, double, triple references in a single message has also been used successfully in the past, trying to geolocate traffic events from Twitter.

Nevertheless, at the time of these contributions, data was not usually provided for replicability purposes, so the authors could only have access to ideas and techniques that could be compared to their proposals.

In conclusion, the paper is sound, is very well written and organized, uses a good and publicly available dataset with adequate annotations with which to perform analyses and comparisons. The observation in the conclusions that the geocoder has a significant influence in the results comes as no surprise, since this has been reported in several of the works I mentioned earlier. It also must be said that some of the ideas and techniques that are employed in the paper have been tried before, including (1) extraction of location indications from the tweet content, (2) analysis of expressions that indicate a location (spatial prepositions), (3) use of gazetteers and addressing databases (not necessarily embedded in a geocoding product), (4) improved reliability of messages in which multiple references appear, (5) using past messages and user relationships in the definition of a tweet's location. I’d only wish that the authors’ literature review reached further back so that they could get to know and recognize some ideas presented earlier in geocoding, geographic information retrieval, toponym resolution, location-based social networks and other subfields.

Response: The authors thank the reviewer for the positive feedback. The authors agree with all points raised by the reviewer. We have widened our literature review to incorporate earlier studies. In light of new understanding, the authors have also refined sections of the literature review, discussion and conclusion to acknowledge existing findings and clarify the contribution of the paper to the research community.

Reviewer #4: - The major concern is that the contributions of this article are not clear. The authors seem to use existing data, existing methods, and existing tools.

Response: The authors thank the reviewer for the feedback. The authors agree that existing methods were used. What the authors show, however, is that by following a different methodological setup, there are able to produce higher precision on a random sample of a geotagged dataset. 

- At the same time the authors claim that they have developed a "new high precision geocoding approach" - The authors should explicitly describe what their approach is and how it is different to existing approaches.

Response: Thank you for your feedback. The authors agree that the wording was misleading. As such, the authors have revised the manuscript and stated that they enhance existing methods. The authors have acknowledged the use of existing approaches and clarified their contribution in the last paragraph of the previous work section under the subsection ‘Tweet’s point of origin’

- The motivation for filtering based on apps is not clear. Sure, sampling based on data source will impact the result because different apps create different user behavior and data. But why is this relevant? And why is this relevant for this paper in which a new geocoding approach is being developed?

Response: The use of different datasets and data source distribution was done to show how location inference models trained on a proportionate distribution of a geotagged dataset may be impacted when applied to a sample of a non-geotagged dataset where the Tweet source distribution differs. 

Our reason to include this analysis was based on the comments from the previous review who stressed out this possible discrepancy in model transferability. The authors found the issue of different data source distribution to be one that has been overlooked when developing location inference models and decided to include this analysis hand in hand with the proposed location inference approach. The authors believe that combining these two aids weight to the paper.

---

## [Decision Letter · Decision Letter 2]

28 Feb 2023

Enhanced geocoding precision for location inference of tweet text using spaCy, Nominatim and Google Maps. A comparative analysis of the influence of data selection.

PONE-D-21-27579R2

Dear Dr. Serere,

We’re pleased to inform you that your manuscript has been judged scientifically suitable for publication and will be formally accepted for publication once it meets all outstanding technical requirements.

Kind regards,

Pierluigi Vellucci

Academic Editor

PLOS ONE

Additional Editor Comments (optional):

Reviewers' comments:

Reviewer's Responses to Questions

**Comments to the Author**

1. If the authors have adequately addressed your comments raised in a previous round of review and you feel that this manuscript is now acceptable for publication, you may indicate that here to bypass the “Comments to the Author” section, enter your conflict of interest statement in the “Confidential to Editor” section, and submit your "Accept" recommendation.

Reviewer #1: All comments have been addressed

Reviewer #3: All comments have been addressed

2. Is the manuscript technically sound, and do the data support the conclusions?

Reviewer #1: Yes

Reviewer #3: Yes

3. Has the statistical analysis been performed appropriately and rigorously? 

Reviewer #1: Yes

Reviewer #3: Yes

4. Have the authors made all data underlying the findings in their manuscript fully available?

Reviewer #1: Yes

Reviewer #3: Yes

5. Is the manuscript presented in an intelligible fashion and written in standard English?

Reviewer #1: Yes

Reviewer #3: Yes

6. Review Comments to the Author

Reviewer #1: The revised version of the manuscript has addressed all the issues in the previous drafts and merits publication. In this reviewer's opinion, it has improved significantly in many aspects, beyond the issues identified by this and other reviewers in previous revisions.

In case the authors find it useful, I include a short list of minor editing errors that can be addressed:

- References without spacing (lines 125, 134)

- "Error! Reference not found" (lines 196, 197, 198, 274, 325, 415, 470)

- Space between "Table 4" and "we" in line 369 (and line 476)

- Informal "~56%" in line 407 (I suggest "around" or "close to")

- Underline in pages 21, and use of bold text is in my experience unusual (use sparsely, e.g. as in the key points in the conclusions)

- Inconsistent decimal precision in table 8 (44.38 with two decimals VS e.g. 81.0, 60, 96.8)

- "third applications" in line 525 has "party" missing

Reviewer #3: (No Response)

7. PLOS authors have the option to publish the peer review history of their article (what does this mean?). If published, this will include your full peer review and any attached files.

Reviewer #1: No

Reviewer #3: No

---

## [Editor Report · Acceptance letter]

2 Mar 2023

PONE-D-21-27579R2 

Enhanced geocoding precision for location inference of tweet text using spaCy, Nominatim and Google Maps. A comparative analysis of the influence of data selection. 

Dear Dr. Serere:

I'm pleased to inform you that your manuscript has been deemed suitable for publication in PLOS ONE. Congratulations! Your manuscript is now with our production department. 

Kind regards, 

on behalf of

Dr. Pierluigi Vellucci 

Academic Editor

PLOS ONE